# A 'hidden problem': Nature, prevalence and factors associated with sexual dysfunction in persons living with HIV/AIDS in Uganda

**Brian Byamah Mutamba**[1], **Godfrey Zari Rukundo**[2]*, **Wilber Sembajjwe**[3], **Noeline Nakasujja**[4], **Harriet Birabwa-Oketcho**[1], **Richard Stephen Mpango**[5,6], **Eugene Kinyanda**[3,5]

**1** Butabika National Referral Mental Hospital, Kampala, Uganda, **2** Department of Psychiatry, Mbarara University of Science and Technology, Mbarara, Uganda, **3** Statistical Section, MRC/UVRI and LSHTM Uganda Research Unit, Entebbe, Uganda, **4** Department of Psychiatry, College of Health Sciences, Makerere University, Kampala, Uganda, **5** Mental Health Section, MRC/UVRI and LSHTM Uganda Research Unit & Senior Wellcome Trust Fellowship, Entebbe, Uganda, **6** Department of Mental Health, School of Health Sciences, Soroti University, Soroti, Uganda

* grukundo@must.ac.ug

**Data Availability Statement:** All relevant data are within the paper.

**Funding:** This study was funded by MRC TA.10.40200.011 core funding to the Mental health

## Abstract

### Background

We conducted a clinic-based cross-sectional survey among 710 people living with HIV/AIDS in stable 'sexual' relationships in central and southwestern Uganda. Although sexual function is rarely discussed due to the private nature of sexual life. Yet, sexual problems may predispose to negative health and social outcomes including marital conflict. Among individuals living with HIV/AIDS, sexual function and dysfunction have hardly been studied especially in sub-Saharan Africa. In this study, we aimed to determine the nature, prevalence and factors associated with sexual dysfunction (SD) among people living with HIV/AIDS (PLWHA) in Uganda.

### Methods

We conducted a clinic based cross sectional survey among 710 PLWHA in stable 'sexual' relationships in central region and southwestern Uganda. We collected data on socio-demographic characteristics (age, highest educational attainment, religion, food security, employment, income level, marital status and socio-economic status); psychiatric problems (major depressive disorder, suicidality and HIV-related neurocognitive impairment); psychosocial factors (maladaptive coping styles, negative life events, social support, resilience, HIV stigma); and clinical factors (CD4 counts, body weight, height, HIV clinical stage, treatment adherence).

### Results

Sexual dysfunction (SD) was more prevalent in women (38.7%) than men (17.6%) and majority (89.3% of men and 66.3% of women) did not seek help for the SD. Among men, being of a religion other than Christianity was significantly associated with SD (OR = 5.30, 95%CI 1.60–17.51, p = 0.006). Among women, older age (> 45 years) (OR = 2.96, 95%CI

project of MRC/UVRI and LSHTM under the headship of Professor Eugene Kinyanda to undertake the 'HIV clinical trials preparedness studies among patients with Severe Mental ILlnEss in HIV endemic Uganda (SMILE Study) The funders had no role in study design, data collection and analysis, decision to publish, or preparation of the manuscript.

**Competing interests:** The authors have declared that no competing interests exist.

**Abbreviations:** FSD, Female sexual disorder; LMICs, Low- and Middle-Income Countries; MSD, Male sexual disorder; PLWHA, People living with HIV/AIDS; SD, Sexual dysfunction; WHO, World Health Organization.

1.82–4.79, p<0.01), being widowed (OR = 1.80, 95%CI 1.03–3.12, p = 0.051) or being separated from the spouse (OR = 1.69, 95% CI 1.09–2.59, p = 0.051) were significantly associated with SD. Depressive symptoms were significantly associated with SD in both men (OR = 0.27, 95%CI 0.74–0.99) and women (OR = 1.61, 95%CI 1.04–2.48, p = 0.032). In women, high CD4 count (OR = 1.42, 95% CI 1–2.01, p = 0.05) was associated with SD.

## Conclusion

Sexual dysfunction has considerable prevalence among PLWHA in Uganda. It is associated with socio-demographic, psychiatric and clinical illness factors. To further improve the quality of life of PLWHA, they should be screened for sexual dysfunction as part of routine assessment.

## Introduction

Normal sexuality is a psycho-biological activity necessary for psychological and biological balance in most humans, and for the continuation of human life. It involves the feelings of desire, behaviour that brings pleasure to oneself and one's partner, and stimulation of the primary sex organs, including coitus. The activity should be devoid of inappropriate feelings of guilt or anxiety, and compulsions [1]. Anatomy, physiology, psychology, the culture in which one lives, relationships with others, and the developmental experiences throughout the life cycle determines sexuality. Satisfying sexual life is a key quality of life concern that is rarely inquired into. Although sexual dysfunction is not well studied, according to available literature, it has a global prevalence of about 40–45% among adult women and 20–30% among adult men and the prevalence increases with age [2, 3]. There are four main domains of sexual disorders depending on the affected aspect: desire, arousal, orgasm and pain associated with sex [4]. The most common factors associated with sexual dysfunction differ according to gender but the shared factors include general health status, diabetes mellitus, cardiovascular disease, genitourinary disease, psychological problems and socio-demographic characteristics [2].

Sexual function in the context of HIV/AIDS has mainly been studied with respect to HIV transmission [5]. From a public health perspective, sexual problems may predispose individuals to unsafe sexual practices as well as intimate partner relationship discord including marital violence, and poor ART adherence. In addition, sexual dysfunction may be a symptom of an undiagnosed mental disorder such as major depressive disorder [6]. Both women and men living with HIV experience common risk factors for sexual dysfunction including poor mental health, hormonal imbalances, pharmacological and other co-morbid conditions and substance abuse [7, 8].

The life-span of people living with HIV/AIDS (PLWHA) has improved significantly due to the availability of highly effective antiretroviral therapy (ART). PLWHA are living longer and healthier lives with less comorbidity due to the availability of effective ART with the majority remaining sexually active for many years. As a result, attention has now shifted to improving quality of life [9]. Previous research in this area has mainly focused on sexual behaviour in the context of assessing HIV transmission risk dynamics, condom use and on the determinants of unintended pregnancy [10–12]. Other aspects of sexual health including sexual dysfunction (SD) and satisfaction with sexual relationships have not been studied as much [13]. Although it was initially thought to be unimportant to this sub-population, the few studies done so far indicate that poor sexual functioning is an important area of concern for PLWHA [14].

According to studies by Rosen [15, 16], sexual dysfunction can be categorised as: i) desire disorders; ii) arousal disorders; iii) orgasm disorders; and iv) pain disorders. Another area of sexual dysfunction is satisfaction with one's sexual relationship. In this study, we aimed to determine the nature, prevalence and factors associated with SD among PLWHA in Uganda.

## Methods

### Study design

We conducted a cross sectional survey among 710 PLWHA at HIV clinics run by The AIDS Support Organisation (TASO) in Entebbe (central region) and Masaka (southwestern) in Uganda. The sample consisted of males and females in stable sexual relationships that consented to participate in this sub-study. This sub-study was undertaken as part of Prof Eugene Kinyanda's Senior EDCTP Fellowship funded study entitled, 'Clinical trials in HIV/AIDS in Africa: Should they routinely control for mental health factors?'

### Study setting

TASO is the oldest indigenous organization that provides HIV/AIDS care and support services in Uganda and Sub-Saharan Africa [17]. It was founded in 1987 and has several branches in many parts of Uganda. TASO offers a comprehensive care package including medical care, psychosocial support, and sensitization about adherence to medication, especially ART. The service of TASO directly complements the efforts of the Ugandan Ministry of Health. Although the TASO centers are located in the district towns, they provide services to clients within a radius of 75 kilometers.

The TASO centre in Masaka was established in 1988 and is located within the Masaka regional referral hospital premises in southwestern Uganda. TASO Masaka serves over 40000 HIV-positive clients, their families and communities. On the other hand, TASO Entebbe commenced work in 1991 and is located within Entebbe Municipal Council in central Uganda. Entebbe being urbanized has an ever-growing and transient population including traders, fish mongers, and uniformed personnel. It is the first and last stop for international tourists. TASO Entebbe serves 7000 clients, their families and surrounding communities.

### Sampling and sample size

This was a cross-sectional study that was part of a prospective cohort of HIV positive adults attending at 2 specialized HIV clinics run by The AIDS Support Organization (TASO) in Uganda: TASO HIV clinic in Masaka and Entebbe (rural and semi-urban). The study enrolled a sample of 710 ART naı̈ve HIV positive adults. Initiation of ART was implemented by TASO independently of the study. At the time of the study, national treatment guidelines for HIV-infected individuals recommended the initiation of ART at a CD4 cell count of below 250 cells/mL. In addition, individuals initiating ART were required to have identified an appropriate treatment supporter.

### Eligibility criteria

Inclusion criteria. i) A person who was living with HIV (PLWH) who was ART naïve and registered with the outpatient clinic at either TASO Entebbe and TASO Masaka clinics; ii) was an adult at the time of enrolment (at least 18 years old); iii) was conversant with English or Luganda, the language into which the research protocols were translated; iv) gave informed consent after adequate explanation of the study objectives, procedures and expected benefits. Exclusion criteria. i) was too sick (those patients who came to the HIV clinics with severe

physical health problems and were unwell and needed emergency assessment and hospitalization)] and ii) unable to understand the study instruments.

## Study variables

A range of explanatory variables that included socio-demographic, psychiatric illness and clinical factors were assessed. The socio-demographic factors included age, highest educational attainment, religion, food security, employment, income level, marital status and socio-economic status (assessed using an index constructed from having common household items and facilities such as electricity, a car, a radio, a bicycle, a telephone, a refrigerator, a lantern and a flask) [18]. Two categories of low and high socio-economic status were generated using principal component analysis. The psychiatric illness factors included major depressive disorder and suicidality (assessed using the Mini International Neuropsychiatric interview (M.I.N.I. Plus) [19]; severity of depressive symptoms was assessed using the Centre for Epidemiological Studies-Depression questionnaire (CES-D) [20]. Psychosocial factors assessed included: Maladaptive Coping Style (MCS) (assessed using the Carver Brief COPE (Coping Orientation to Problems Experienced Inventory) [21], negative life events (assessed using the adverse life events module of the European Parasuicide Interview Schedule modified for the Ugandan situation [22], Social support (assessed using the Multidimensional Scale of Perceived Social Support (MSPSS) [23], resilience (assessed using the Connor-Davidson Resilience Scale (CD-RISC) [24] and HIV stigma (assessed using the brief HIV Stigma Scale) [25]. Clinical factors assessed included: CD4 nadir, CD4 counts in the last 6 months, body weight, height and HIV clinical stage (assessed using the WHO clinical staging criteria [26] and treatment adherence.

Sexual behaviour differs between males and females and hence two different instruments were used to assess for the dependent variable of sexual dysfunction. Among males, SD was assessed using the IIEF [27]. The IIEF is a 15-item

questionnaire that was developed by Rosen and colleagues (1997) for use in heterosexual men to assess five domains of male sexual function, including: desire (items 11 and 12), erectile function (items 1–5 and 15), orgasmic function (items 9 and 10), intercourse satisfaction (items 6, 7 and 8) and overall satisfaction (items 13 and 14) [25]. It has questions such as, *'How often were you able to get an erection during sexual activity?'* Responses are scored on a 6-point Likert scale with items: 0 = No sexual activity; 1 = Almost never or never; 2 = A few times (less than half the time); 3 = Sometimes (about half the time); 4 = Most times (more than half the times); 5 = Almost always or always. It has been linguistically validated in 32 languages including in South African dialects [27]. In this study responses were restricted to the period covering the last 4 weeks.

The primary outcome measure for sexual dysfunction among men was erectile dysfunction (defined as a cut-off score of 25 or less on the erectile function domain of the IIEF for a man in a stable relationship) [27]. The other 4 domains of the IIEF and the total score on the IIEF were considered as secondary outcomes among men in this study.

Sexual dysfunction in women was assessed using the FSFI a tool that was developed by Rosen and colleagues]. The FSFI is a 19-item questionnaire which among western respondents has been found to consist of 6 domains, namely: desire (items 1 and 2), subjective arousal (items 3–6), lubrication (items 7–9), orgasm (items 11–13), global satisfaction (items 14–16) and pain (items 17–19) [28]. It consists of question items such as, *'Over the past 4 weeks, how often did you feel sexual desire or interest?'*. Responses are scored on a 5-point Likert scale with items: 1 = Almost never or never; 2 = A few times (less than half the time); 3 = Sometimes (about half the time); 4 = Most times (more than half the times); 5 = Almost always or always. In this study responses were restricted to the period covering the last 4 weeks (last month).

The primary outcome measure for sexual dysfunction among women was the total FSFI score (where a cut-off score of 26.55 or less is regarded as suggestive of sexual dysfunction) [29]. Weigel and colleagues in cross-validation studies of the FSFI have recommended a total cut-off score of 26.55 for differentiating females with or without sexual dysfunction [29].

Since both the IIEF and the FSFI were used in this socio-cultural setting for the first time, they were taken through a local translation process from English to Luganda (the predominant language spoken in central and south-western region) and subsequent determination of their internal consistency (assessed using the Cronbach's alpha). The local translation process involved forward and back translation of each of these instruments by two independent teams of mental health professionals. The teams selected to undertake this process were conversant with both English and Luganda. The teams then met for a consensus meeting in which the final back-translated English version was compared to the initial English version to assess for consistency. A consensus meeting was then held to address those items where there were wide variations in the two English versions. In this study, the internal consistency of the IIEF was 0.96 and that of the FSFI was 0.97 (a value greater than 0.70 is considered adequate). Trained study psychiatric nurses administered the study tools to the study participants.

## Statistical analysis

Statistical analyses were conducted using Stata release 15 (Stata Corp, TX, USA). Analyses were carried out separately for males and females. Frequencies to the FSFI and IIEF were estimated with their percentages. Categorical variables were described using frequencies and percentages while continuous variables were described using median (IQR). Health seeking behaviour of the respondents was also described using frequencies and percentages. Two binary outcome variables "sexual dysfunction for men and sexual dysfunction for women" were generated. To investigate the association of sexual dysfunction with socio-demographic and psychosocial and psychiatric illness factors, a two-step procedure was adopted. social economic status was categorized into categories from a list of household items using Principal Component Analysis. During the first step, bivariate associations between each of the outcome variables and the independent variables were assessed using simple logistic regression models.

In the second step, those socio-demographic, psychosocial and psychiatric illness factors that attained a level of significance of P ≤ 0.2 (liberal cut-off point) with bivariate analyses were included in the final multivariate logistic regression models that assessed for the factors that were independently significantly associated with sexual dysfunction. We were not able to conduct subgroup analyses by homo- and heterosexual respondents. Education was categorized as no formal education, primary, secondary and tertiary. Religion was also categorised into 2 categories i.e. 'christian' and 'muslim & others'. All analyses adjusted for age.

## Results

### Participant characteristics

A total of 159 men and 551 women participated in the study. The small number of men in the study could have affected the results. The majority of the participants were married and they belonged to the Christian faith (Table 1). The respondents were females and the age of the participants ranged between the age 18 and 65 years.

### Nature and prevalence of sexual dysfunction

Low sexual desire, reduced levels of sexual arousal, not reaching orgasm and experiencing pain or discomfort were the prevalent problems among women. A considerable number of women

**Table 1. Characteristics of study participants on key socio-demographic variables (should not include analyses of associated factors).**

| Variable | Level | Women N = 551(%) | Men N = 159(%) |
|---|---|---|---|
| Highest education attained | No formal education | 69(12.5) | 30(18.9) |
| | Primary | 328(59.5) | 80(50.3) |
| | Secondary | 139(25.2) | 20(12.6 |
| | Tertiary | 14(2.5) | 28(17.6) |
| Religion | Christians | 469(85.1) | 146(91.8) |
| | Muslims & others | 82(14.9) | 13(8.2) |
| Age | Median (IQR) | 33(20; 57) | 35 (21; 57) |
| Food security status | Yes | 445(80.8) | 133(83.6) |
| | No | 104(18.9) | 26(16.4) |
| Employment status | Farmer | 159(28.9) | 50(31.4) |
| | Professionals | 16(2.9) | 12(7.5) |
| | Self-employed/Business | 202(36.7) | 48(30.2) |
| | Unemployed | 163(29.6) | 32(20.1) |
| Socio economic status (SES) | Low SES | 228(41.4) | 36(22.6) |
| | High SES | 321(58.3) | 123(77.4) |
| Marital status | Currently married | 273(49.5) | 102(64.1) |
| | Widowed | 90(16.3) | 10(6.3) |
| | Separated | 137(24.9) | 30(18.9) |
| | Single | 50(9.1) | 17(10.7) |

($\geq$35%) did not engage in sexual activity in the previous month compared to men ($<$ 20%). Among men; low levels of sexual desire and number of times sexual intercourse was attempted were the prevalent problems. The number of times sexual intercourse was attempted was measured as" no attempts, one to two attempts/ three to four attempts, five to six attempts, seven to ten attempts/ Eleven or more attempts". In this case, fewer sexual attempts were assumed to indicate lower sex drive and, therefore sexual dysfunction.

Overall, the prevalence of sexual dysfunction was 17.6% (15.1 to 19.1) and 38.7% (35.9 to 40.1) in men and women respectively, indicating that in PLWHA, women are more affected than men. The majority of both men (89.3%) and women (66.3%) did not seek help for sexual dysfunction, but more women (25%) sought help from a health worker compared to men (7.1%)). Of note, 3.4% and 6.6% of women sought help from a traditional or faith healer respectively compared to 0% and 3.6% in men (Tables 2 and 3).

Some of the factors met the apriori cut-off and were statistically significant at bivariate analysis (Table 4). However, they presented multicollinearity in the multivariate model and therefore reducing precision of the estimates and were therefore eliminated.

## Factors associated with sexual dysfunction

With respect to women; age was significantly associated with sexual dysfunction, with women aged 45 years and above almost three times more likely to have the problem. A marital status indicated as widowed or separated was a risk factor for SD in women (Table 5). Employment status was found to be associated with sexual dysfunction. Major depressive disorder was the only psychological/social factor significantly associated with sexual dysfunction (1.61 (1.04–2.48), p = 0.032) in women. With respect to HIV-related factors, a high CD4 count ($\geq$ 500) was the only one with a statistically significant association to sexual dysfunction (1.42 (1–2.01; p = 0.05) (Table 5). Stroke was also found highly associated with sexual dysfunction.

With respect to men, religion (being Muslim or other faith) and age were the only socio-demographic factors significantly associated with erectile dysfunction. Severity of depressive

**Table 2. Prevalence of sexual dysfunctions among women living with HIV/AIDS in Uganda.**

| Disorder | Level | Frequencies n (%) |
|---|---|---|
| Over the last four weeks | | |
| Sexual Desire problems | | |
| How often did you feel sexual desire or interest? (n = 551) | Almost never or never | 147(26.7) |
| How would you rate your level (degree) of sexual desire or interest? (n = 548) | Very low or none at all | 135(24.5) |
| *Sexual Desire problems (at least one positive item from items 1 to 2) N = 551* | **Positive** | **176(31.9%)** |
| Number of Female that are sexually active*N = 551 | **Yes** | **345(62.6%)** |
| Sexual Arousal problems* n = 345 | | |
| How often did you feel aroused ("turned on") during sexual activity or intercourse? | Almost never or never | 48(13.9%) |
| How would you rate your level of sexual arousal ("turn on") during sexual activity or intercourse? | Very low or none at all | 41(11.9%) |
| How confident were you about becoming sexually aroused during sexual activity or intercourse? | Very low or no confidence | 45(13.0%) |
| How often have you been satisfied with your arousal (excitement) during sexual activity or intercourse? | Almost never or never | 26(7.5%) |
| *Sexual Arousal problems* (at least one positive item from items 3 to 6) | **Positive** | **99(28.7%)** |
| Sexual Lubrication problems* n = 345 | | |
| How often have you become lubricated ("wet") during sexual activity or intercourse? | Almost never or never | 26(7.5%) |
| How difficult was it to become lubricated ("wet") during sexual activity or intercourse? | Extremely /very difficult or impossible | 31(8.9%) |
| How often did you maintain your lubrication ("wetness") until completion sexual activity or intercourse | Almost never or never | 22(6.4%) |
| How difficult was it to maintain your lubrication ("wetness") until completion sexual activity or intercourse | Extremely /very difficult or impossible | 28(8.1%) |
| *Sexual Lubrication problems* (at least one positive item from items 7 to 10) | Positive | 45(13.0%) |
| Orgasm problems* n = 345 | | |
| When you had sexual stimulation or intercourse How often did you reach orgasm (climax)? | Almost never or never | 46(13.3%) |
| When you had sexual stimulation or intercourse how difficult was it for you to reach orgasm (climax)? | Extremely /very difficult or impossible | 45(13.0%) |
| How satisfied were you with your ability to reach (climax) orgasm during sexual activity or intercourse? | Moderately and Very dissatisfied | 100(28.9%) |
| *Orgasm problems* (at least one positive item from items 11 to 13) | Positive | 115(33.3%) |
| Sexual Pain problems* n = 345 | | |
| How often did you experience discomfort or pain following vaginal penetration? | Almost always or Most times | 105(30.4%) |
| How often did you experience discomfort or pain during vaginal penetration | Almost always or Most times | 37(10.7%) |
| How would you rate your level (degree) of discomfort or pain during vaginal penetration? | High | 36(10.4%) |
| *Sexual Pain problems* (at least one positive item from items 14 to 16) * | Positive | 139(40.3%) |
| *Sexual dysfunction (at least one positive item from item 1 to 13 and then 17 to 19)* | Positive | 348(63.2%) |
| Dissatisfaction with sexual experience with current partner** n = 551 | | |

*(Continued)*

**Table 2.** (Continued)

| Disorder | Level | Frequencies n (%) |
|---|---|---|
| How satisfied have you been with the amount of emotional closeness during sexual activity or intercourse between you and your partner? | Moderately and Very dissatisfied | 78(14.2%) |
| How satisfied have you been with sexual relationship with your partner? | Moderately and Very dissatisfied | 193(35.0%) |
| How satisfied have you been with your overall sexual life? | Moderately and Very dissatisfied | 192(34.8%) |
| Dissatisfaction with sexual experience with current partner (at least one positive from items 14 and 15) | Positive | 214(38.8%) |

Note *Percentage among sexually active

**Percentage among those in an intimate partner relationship

symptoms was the only psychological and social factor significantly associated with ED. Stroke was also found highly associated with erectile dysfunction.

No HIV-related or anthropometric factors were significantly associated with ED.

## Discussion

We sought to explore the topic of sexual functioning among patients living with HIV in Uganda, a rather taboo topic in this setting, with an aim of informing quality of care initiatives. SD is a relatively common problem experienced by men and women living with HIV in Uganda, with one in five men and two in five women affected. Studies on SD in PLWHA in similar contexts is scarce, however some of our findings compare with those in the extant literature. It is worth noting that for both men and women with SD, majority did not seek treatment. This was especially so for men, with less than 1 in 10 seeking any form of help. Understanding why women seek help from alternate care providers (traditional and faith healers), and what help is received is an area for further inquiry.

### Nature and prevalence of dysfunction among PLWHA

We found that SD was more prevalent in women (38.7%) compared to men (17.6%). However, there was disproportionate representation of women in the sample. It is possible that this could have led to the difference in the burden of SD. Our findings are similar to what has been reported in previous studies [30–33] which have also found high prevalence of sexual dysfunction among women than in men. Unfortunately, SD is a burden that is rarely discussed in clinical encounters and therefore often goes unattended. According to our study findings, 89.3% of men and 66.3% of women did not seek help for the dysfunction. This finding is similar to what has been reported in previous studies which found that the healthcare system may not be prepared for the immediate and continuing support for sexual complaints [34–37].

### Factors associated with SD in HIV-infected men

Poor erectile function is thought to be due to HIV associated co-morbid physical conditions particularly when it manifests among HIV-infected men without AIDS [5, 38]. This is not consistent with our findings that no HIV-related physical or anthropometric factors were associated with SD. Studies from elsewhere have identified a range of physical factors associated with SD in men; hypogonadism and low serum testosterone levels were the more commonly reported factors [8, 39]. Other physical conditions that have been reported to be associated

**Table 3. Prevalence of sexual dysfunctions among men living with HIV/AIDS in Uganda.**

| Disorder | Level | Frequencies (%) |
|---|---|---|
| Over the last four weeks . . . . . .. | | |
| Sexual Desire problems n = 157 | | |
| How often have you felt sexual desire? | Almost never/ A few times | 54(34.4%) |
| How would you rate your level of sexual desire? | Very low/ Low | 39(24.8%) |
| *Sexual Desire problems (at least one positive item from items 11 to 12)* | Positive | 62(39.5%) |
| Number of Male that are sexually active | Yes | 139(88.5%) |
| Sexual Arousal problems* n = 139 | | |
| How often were you able to get an erection during sexual activity | Almost never/ a few times | 39(28.1%) |
| When you had erections with sexual stimulation, how often were your erections hard enough for penetration? | Almost never/ A few times | 37(26.6%) |
| When you attempted intercourse, how often were you bale to penetrate (enter) your partner? | Almost never/ A few times | 30(21.6%) |
| During sexual intercourse, how often were you able to maintain your erection after you had penetrated (entered) your partner? | Almost never/ A few times | 33(23.7%) |
| During sexual intercourse, how difficult was it to maintain your erection to completion of intercourse? | Extremely difficult/ Very difficult | 5(3.6%) |
| How do you rate your confidence that you could get and keep an erection? | Very Low/ Low | 33(23.7%) |
| *Sexual Arousal problems (at least one positive item from items 1 to 5 and item 15)* | Positive | 84(60.4%) |
| Ejaculation problems | | |
| When you had sexual stimulation <u>or</u> intercourse, how often did you ejaculate? | Almost never/ A few times | 16(11.5%) |
| When you had sexual stimulation or intercourse, how often did you have the feeling of orgasm or climax? | Almost never/ A few times | 42(30.2%) |
| *Ejaculation problems (at least one positive item from items 9 to 10)* | Positive | 49(35.2%) |
| *Sexual dysfunction (at least one positive item from items: 11,12, 1 to 5, 15, 9,10)* | Positive | 102(73.4%) |
| Dissatisfaction with Sexual experience | | |
| When you attempted sexual intercourse, how often was it satisfactory for you? | Almost never/ A few times | 18(12.9%) |
| How much have you enjoyed sexual intercourse? | No enjoyment at all/ Not very enjoyable | 9(6.5%) |
| How satisfied have you been with your overall sex life? | Very dissatisfied/ Moderately dissatisfied | 35(25.2%) |
| How satisfied have you been with your sexual relationship with your partner? | Very dissatisfied/ Moderately dissatisfied | 30(21.6%) |
| *Dissatisfaction with Sexual experience (at least one positive item from items 7,8,13,14)* | Positive | 36.7% |

**Note** *Percentage among sexually active

with SD in HIV-infected individuals include hepatopathy, diabetes mellitus, hyperlipidaemia, hypertension, vascular disease, and alcohol dependence [38, 40]. Similar to previous studies, in our study, the physical conditions were significantly associated with SD in men were stroke and diabetes. The two disease conditions have known vascular complications that often cause sexual dysfunction. However, the number of men was smaller than that of women in this study. This could have affected the results.

Psychological or mental health factors are thought to play a major role in sexual dysfunction. Feelings of guilt from having acquired HIV via sexual transmission has been reported to

**Table 4. Factors associated to sexual dysfunction at Bivariate (unadjusted) analysis.**

| Variable | Level | Women Crude odds ratio (95% CI) | Men Crude odds ratio (95% CI) |
|---|---|---|---|
| Highest education attained | | P = 0.422 | P = 0.819 |
| | No formal education | 1 | 1 |
| | Primary | 0.83(0.49–1.41) | 0.90(0.37–2.21) |
| | Secondary | 0.66(0.38–1.20) | 0.36(0.18–1.20) |
| | Tertiary | 1.3(0.41–4.11) | 1.3(0.41–4.11) |
| Age | | **P<0.001** | **P<0.001** |
| | 18–44 years | 1 | 1 |
| | 45 years and above | 2.96(1.82–4.79) | 3.96(1.02–6.31) |
| Religion | | P = 0.194 | P = 0.009 |
| | Christians | 1 | 1 |
| | Muslims & Others | 1.37(0.85–2.20) | 4.83(1.48–15.74) |
| Food security status | | P = 0.450 | P = 0.74 |
| | Yes | 1.18(0.76–1.85) | 1.21(0.38–3.84) |
| | No | 1 | 1 |
| Employment status | | P = 0.03 | P = 0.66 |
| | Farmer | 1 | 1 |
| | Professionals | 0.96(0.34–2.72) | 1.05(0.19–5.72) |
| | Self-employed/Business | 0.86(0.57–1.32) | 1.56(0.57–4.30) |
| | Unemployed | 0.52(0.33–0.82) | 0.75(0.2–2.73) |
| Socio economic status | | P = 0.157 | P = 0.409 |
| | Low ses | 1 | 1 |
| | High ses | 0.78(0.55–1.10) | 0.68(0.27–1.71) |
| Marital status | | **P<0.001** | P = 0.70 |
| | Currently married | 1 | 1 |
| | Widowed | 2.64(1.62–4.30) | 0.52(0.62–4.35) |
| | Separated | 1.78(1.16–2.71) | 1.42(0.52–3.81) |
| | Single | 1.14(0.60–2.16) | 0.62(0.13–3.0) |
| Major Depressive disorder | | **P = 0.033** | P = 0.81 |
| | Yes | 1.59(1.03–2.43) | 1.14(0.39–3.3) |
| | no | 1 | 1 |
| Suicidality | | P = 0.902 | P = 0.481 |
| | Yes | 1.04(0.57–1.90) | 0.50(0.06–4.1) |
| | no | 1 | 1 |
| Severity of depressive symptoms | | P = 0.151 | P = 0.073 |
| | 0 symptoms | 1 | 1 |
| | 1–10 symptoms | 0.66(0.39–1.10) | 0.31(0.11–0.86) |
| | 11–20 symptoms | 0.57(0.32–1.01) | 0.28(0.78–1.02) |
| Negative life events | | P = 0.412 | P = 0.368 |
| | 0–5 events | 1 | 1 |
| | 6–10 events | 0.78(0.53–1.15) | 0.68(0.24–1.97) |
| | 11–17 events | 0.81(0.43–1.51) | 2.06(0.57–7.39) |
| Social support | | P = 0.406 | P = 0.019 |
| | Yes | 1 | 1 |
| | No | 1.18(0.79–1.76) | 0.35(0.15–0.83) |
| HIV neurocognitive impairment | | P = 0.503 | - |
| | Have dementia | 1.15(0.76–1.73) | |
| | No dementia | 1 | |

*(Continued)*

**Table 4.** (Continued)

| Variable | Level | Women Crude odds ratio (95% CI) | Men Crude odds ratio (95% CI) |
|---|---|---|---|
| weight | | P = 0.386 | P = 0.913 |
| | > = 61 kgs | 1 | 1 |
| | 30–60 kgs | 0.86(0.60–1.22) | 0.94(0.40–2.22) |
| CD4 count | | **P = 0.025** | P = 0.368 |
| | <500 | 1 | 1 |
| | > = 500 | 1.48(1.05–2.09) | 0.65(0.26–1.67) |
| Adherence to HIV medication | | P = 0.783 | P = 0.234 |
| | Yes | 1.06(0.69–1.62) | 1.23(0.42; 1.49) |
| | No | 1 | 1 |
| Risky sexual behaviour | | P = 0.732 | P = 0.433 |
| | Yes | 0.91(0.54–1.54) | 0.98(0.76; 1.97) |
| | No | 1 | 1 |
| Diabetes | | P = 0.294 | P = 0.043 |
| | Yes | 2.4(0.39–14.5) | 1.99(1.01; 2.13) |
| | No | 1 | 1 |
| stroke | | **P = 0.001** | **P = 0.002** |
| | Yes | 2.19(1.15–6.43) | 1.94(1.01; 5.21) |
| | No | 1 | 1 |

* HIV neurocognitive impairment among men was not included due to perfect collinearity.

negatively influence sexual response in men [40]. This may explain the significant association between religion and SD in men. However, this has not been documented elsewhere. In keeping with the evidence, severity of depression was associated with SD in men. Depression and anxiety are some of the most important mental health factors associated with sexual dysfunctions [8, 39]. In a study of men infected with HIV, those who reported SD had significantly higher scores of psychological symptoms although it was not clear whether the psychological symptoms were a cause or an effect of altered sexual function [39, 41].

Sexual function can negatively be impacted by the effects of HIV treatment [42]. In one study by Lallemand and colleagues (2002), SD was found to be prevalent in HIV-infected men receiving ART irrespective of whether the treatment regimen contained protease inhibitors (PIs) or not. In that study the most prevalent forms of SD were loss of libido, erectile dysfunction and orgasmic disorders [14]. Though low level of sexual desire was one of two prevalent problems among men in our study, it was not found to be associated with ART adherence. A number of studies on the other hand have reported that SD is more common in people taking Protease Inhibitors [38, 43–45]. Other non-antiretroviral factors that have been associated with SD in HIV-infected men include age, sexual preference, duration of HIV infection, CD4 T-cell nadir and the occurrence of opportunistic infection [46].

### Factors associated with SD in HIV-infected women

Our study found a prevalence of FSD (38.7%) which is comparable to 32% in a study conducted among HIV-positive women on ART in Italy [47] but much lower than a prevalence of 61% among HIV-positive women in Nigeria [48]. These are higher than the prevalence of 25% found in study of HIV-positive women in Europe [49] indicating the wide-ranging variability in the prevalence of FSD among HIV-positive women across the globe. This could be as a result of contextual and cultural factors related to experience and disclosure about SD [48].

**Table 5. Factors associated with sexual dysfunction (Multivariate model).**

| Factor | Level | Women aOR (95%CI) | Men aOR (95%CI) |
|---|---|---|---|
| Age | | **P<0.001***  | **P<0.001***  |
| | 18–44 years | 1 | 1 |
| | 45 years and above | 2.92(1.78–4.79) | 3.90(1.02–6.01) |
| Religion | | P = 0.194 | **P = 0.009***  |
| | Christians | 1 | 1 |
| | Muslims & Others | 1.40(0.85–2.20) | 4.83(1.50–15.74) |
| Employment status | | **P = 0.030***  | P = 0.461 |
| | Farmer | 1 | 1 |
| | Professionals | 0.96(0.34–2.72) | 1.05(0.19–5.72) |
| | Self-employed/Business | 0.86(0.57–1.32) | 1.56(0.57–4.30) |
| | Unemployed | 0.52(0.33–0.82) | 0.75(0.2–2.73) |
| Marital status | | **P<0.001***  | P = 0.120 |
| | Currently married | 1 | 1 |
| | Widowed | 2.64(1.62–4.30) | 0.52(0.62–4.35) |
| | Separated | 1.78(1.16–2.71) | 1.42(0.52–3.81) |
| | Single | 1.14(0.60–2.16) | 0.62(0.13–3.0) |
| Major Depressive disorder | | **P = 0.032***  | - |
| | Yes | 1.61 (1.04–2.48) | |
| | no | 1 | |
| Severity of depressive symptoms | | - | P = 0.050 |
| | 0 symptoms | | 1 |
| | 1–10 symptoms | | 0.29(0.10–0.83) |
| | 11–20 symptoms | | 0.27(0.74–0.99) |
| CD4 count | | P = 0.05 | - |
| | <500 | 1 | |
| | > = 500 | 1.42(1–2.01) | |
| Stroke | | **P = 0.016***  | **P = 0.023***  |
| | Yes | 2.89(1.95–8.81) | 2.54(1.03; 6.61) |
| | No | 1 | 1 |
| Social support | | - | P = 0.065 |
| | Yes | | 1 |
| | No | | 0.37(0.15–0.89) |

*Significant at 5% level of significance

Wilson and colleagues reported a higher burden of sexual problems among HIV-positive women compared to HIV negative women, indicating a clear link between HIV infection and sexual problems [13]. In our study, a number of sexual problems, relating to reduced desire for and difficulties during sexual activity, were experienced by women. These findings are similar to findings in HIV-positive women in Nigeria [50]. Reduced sexual desire in women after testing positive for HIV has been reported to be associated with fear of HIV transmission, depression, stress and interpersonal problems [51]. Concerns about HIV transmission, HIV treatment including ART and its side effects may have deleterious effects on female sexual functioning as is the use of alcohol and street drugs [7, 51]. This could be a possible explanation why one in three women who participated in the study did not engage in sexual activity in the month prior.

We found that older age (>45 years) and being widowed or separated were associated with an increased risk of FSD. Oyedokun and others found no association between FSD and religion among HIV-positive women in Nigeria, however, increasing age [47], was significantly associated with FSD [52]. These findings are corroborated in our study. It has been observed that self-perceived body image is a major determinant of FSD [52] and one can argue that increasing or an older age is a contributory factor. Unlike in their study, we did not find education level to be significantly associated with FSD [52].

Consistent with our findings where we did not find any significant association between FSD and ART, most studies that have explored the association between ART and female sexual functioning have not shown any significant association [52–54]. Comparisons of HIV-positive women who were ART naïve and those who had been treated found no difference in sexual functioning [53]. In another study HIV-positive women described diminished sexual activity, a loss of sexual interest and decreased feelings of attractiveness, both before and after the advent of ART [55].

Similar to our findings that CD4 cell count was the only HIV-related factor significantly associated with FSD, a study by Wilson and others (2010) also showed that those with CD4 count of $\leq$199 cells/mm$^3$ had more impaired sexual functioning as assessed by the female sexual function index (FSFI) as compared with those with CD4 counts of 200 cells/mm$^3$ or higher [13]. This was also similar to findings by Oyedokun and others in their study on HIV-positive women in Nigeria [47]. It is possible that that when an individual has low CD4 count they may be more focused on their other physical health problems, with less interest in sex.

Psychosocial variables such as mental health status, quality of life, but not clinical status (illness stage), have been reported to be the main determinants of sexual impairment among HIV-positive women [54, 56]. Depression was the only psychosocial factor associated with FSD in our study, a finding similar to that in other studies of FSD in HIV-positive women in which psychological distress was a significant risk factor for FSD [52, 53]. In another study, women with greater depressive symptoms and greater intrusive thoughts reported poorer sexual quality-of-life with depressive symptoms mediating the association between intrusive thoughts and lower sexual quality-of-life [57]. On the other hand, according to our data and previous studies, age is also associated with sexual dysfunction among men.

## Limitations

This was a cross-sectional study thus it was difficult to establish the direction of causality. We did not study SD in non-HIV-positive men and women and are therefore not able to compare our findings with this population to further understand the role of HIV in SD. In addition, we did we have a qualitative component to further understand sexual behavior and experiences of the respondents. We were not able to conduct subgroup analyses by homo- and heterosexual respondents. There could have been differences according to sexual orientation.

## Conclusion and recommendations

SD is prevalent among PLWHA in Uganda and it is associated with socio-demographic, psychiatric and clinical illness factors. Underreporting and Lack of Seeking Help: A significant percentage of individuals experiencing SD did not seek help for their condition. This underscores the need for increased awareness and open discussions about SD in clinical encounters, as well as improved support systems for those affected.

Factors Associated with SD: Among HIV-infected men, no HIV-related physical or anthropometric factors were associated with SD. However, stroke and diabetes showed a significant association. In contrast, psychological factors, such as religion and severity of depression,

played a role in SD among men. For women, older age (>45 years) and being widowed or separated were associated with an increased risk of female sexual dysfunction (FSD). CD4 cell count was the only HIV-related factor significantly associated with FSD. Depression emerged as a significant psychosocial factor linked to FSD.

Routine assessments and targeted, culturally appropriate interventions for SD should be provided as part of the integrated mental health services in HIV care.

Implications to LMIC

LMICs can benefit from raising awareness about sexual dysfunction among people living with HIV/AIDS (PLWHA). Health authorities and organizations should prioritize educational campaigns aimed at healthcare providers, PLWHA, and the general population to reduce stigma and encourage open discussions about sexual health.

## Author Contributions

**Conceptualization:** Brian Byamah Mutamba, Godfrey Zari Rukundo, Noeline Nakasujja, Harriet Birabwa-Oketcho, Richard Stephen Mpango, Eugene Kinyanda.

**Data curation:** Wilber Sembajjwe.

**Formal analysis:** Wilber Sembajjwe.

**Funding acquisition:** Eugene Kinyanda.

**Investigation:** Godfrey Zari Rukundo, Richard Stephen Mpango, Eugene Kinyanda.

**Methodology:** Godfrey Zari Rukundo, Noeline Nakasujja, Harriet Birabwa-Oketcho, Richard Stephen Mpango, Eugene Kinyanda.

**Project administration:** Richard Stephen Mpango.

**Resources:** Eugene Kinyanda.

**Software:** Wilber Sembajjwe.

**Writing – original draft:** Brian Byamah Mutamba.

**Writing – review & editing:** Brian Byamah Mutamba, Godfrey Zari Rukundo, Wilber Sembajjwe, Noeline Nakasujja, Harriet Birabwa-Oketcho, Richard Stephen Mpango, Eugene Kinyanda.

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
