## [Decision Letter · Decision Letter 0]

9 Nov 2022

PONE-D-22-24725A ‘hidden problem’: Nature, prevalence and factors associated with sexual dysfunction in persons living with HIV/AIDS in UgandaPLOS ONE

Dear Dr. Rukundo,

Thank you for submitting your manuscript to PLOS ONE. After careful consideration, we feel that it has merit but does not fully meet PLOS ONE’s publication criteria as it currently stands. Therefore, we invite you to submit a revised version of the manuscript that addresses the points raised during the review process. Please peruse the reviewers comments and adjust the manuscript accordingly. 

We look forward to receiving your revised manuscript.

Kind regards,

Margaret Williams, Ph.D

Academic Editor

PLOS ONE

Journal Requirements:

“The authors extend appreciation to the Medical Research Council, Uganda (MRC, Uganda) for funding and facilitating the study.”

6. Please include a copy of Table 5 which you refer to in your text on page 18.

7. We note you have included a table to which you do not refer in the text of your manuscript. Please ensure that you refer to Table 4 in your text; if accepted, production will need this reference to link the reader to the Table.

Reviewers' comments:

Reviewer's Responses to Questions

**Comments to the Author**

1. Is the manuscript technically sound, and do the data support the conclusions?

Reviewer #1: Partly

2. Has the statistical analysis been performed appropriately and rigorously? 

Reviewer #1: I Don't Know

3. Have the authors made all data underlying the findings in their manuscript fully available?

Reviewer #1: Yes

4. Is the manuscript presented in an intelligible fashion and written in standard English?

Reviewer #1: No

5. Review Comments to the Author

Reviewer #1: Dear authors, I am providing specific comments in the assigned space, but I am also attaching these in a PDF file, should these be easier to work with.

General

Abstract, sentence starting with ‘Among individuals living with HIV/AIDS…’: Suggest changing to ‘…have hardly been studied, especially in sub-Saharan Africa…’

Hyphenation is a recurring problem in the manuscript:

• Recommend changing first sentence of Abstract to read ‘We conducted a clinic-based cross-sectional survey among 710 PLWHA in stable ‘sexual’ relationships in central and southwestern Uganda.

• Recommend changing ‘HIV related’ to ‘HIV-related’ in the Abstract, Methods section and throughout the manuscript.

• Please change ‘HIV infected’ to ‘HIV-infected’ throughout the manuscript.

• Please change ‘HIV positive’ to ‘HIV-positive’ throughout the manuscript.

Abstract, Methods section: Suggest changing to ‘…psychosocial factors (maladaptive coping styles, negative life events, social support, resilience, HIV stigma)…’

Inappropriate capitalisation if a recurring problem in the manuscript:

• Recommend changing sentence in Abstract, Methods section to ‘…and clinical factors (CD4 counts, body weight, height, HIV clinical stage, treatment adherence).’

• Recommend changing to ‘negative life events’, ‘social support’, and ‘resilience’ in Methods, Study variables, fourth sentence.

• Introduction, third paragraph, fifth sentence: Categories of sexual dysfunction do not need to be capitalised.

• Methods, Study variables, fifth sentence: Recommend changing to ‘body weight’ and ‘height’.

• Table 1: ‘Others’ in ‘Moslems & Others’ don’t need to be capitalised.

• Table 2a, Orgasm problems, second item: ‘How’ does not need to be capitalised.

• Results, Factors associated with sexual dysfunction, first paragraph, third sentence: ‘Depressive’ in ‘Major Depressive disorder’ does not need to be capitalised.

Abstract, first sentence of Results: Suggest changing to ‘SD was more prevalent in women (38.7%) than men (17.6%) and the majority (89.3% of men and 66.3% of women)…’

Abstract, first sentence of Conclusion: Suggest changing to ‘Sexual dysfunction has considerable prevalence among PLWHA in Uganda. It is associated with socio-demographic, psychiatric and clinical illness factors. To further improve the quality of life of PLWHA, they should be screened for sexual dysfunction as part of routine assessment.’

Introduction, first paragraph, second sentence: Suggest changing to ‘As a result, attention has now shifted to…’

Introduction, first paragraph, fourth sentence: Suggest changing to ‘Sexual function in the context of HIV/AIDS has mainly been studied with respect to…’

Introduction, first paragraph, fifth sentence: Suggest changing to ‘From a public health perspective, sexual problems may predispose individuals to unsafe sexual practices as well as intimate partner relationship discord, including marital violence, and poor ART adherence.

Introduction, first paragraph, seventh sentence: Recommend changing ‘ill mental health’ to ‘poor mental health’.

Introduction, second paragraph, second sentence: Suggest changing to ‘It involves the feelings of desire, behavior that brings pleasure to oneself and one’s partner, and stimulation of the primary sex organs, including coitus. The activity should be devoid of…’

Introduction, second paragraph, third sentence: Suggest changing to ‘Anatomy, physiology, psychology, the culture in which one lives, relationships with others, and the developmental experiences throughout the life cycle determines sexuality.’

Introduction, third paragraph, third sentence: Please replace ‘FSD’ with ‘SD’.

Methods, Study setting, second paragraph, third sentence: Suggest changing to ‘On the other hand, TASO Entebbe commenced its work in 1991 and…’

Methods, Study setting, second paragraph, fourth sentence: Suggest changing to ‘Entebbe, being urbanized, has an ever-growing and transient population including traders, fish-mongers, and uniformed personnel. It is also a first and last stop for international tourists.’

Methods, Study variables, first paragraph, first sentence: Suggest changing to ‘A range of explanatory variables that included socio-demographic, psychiatric illness, and clinical factors were assessed.’

Methods, Study variables, first paragraph, second sentence: Suggest changing ‘…constructed from common household items…’ to ‘…constructed from having common household items…’

Methods, Study variables, first paragraph, third sentence: Suggest changing to ‘The psychiatric illness factors included major depressive disorder and suicidality (assessed using the Mini International Neuropsychiatric interview (M.I.N.I. Plus) (Sheehan and Lecrubier 2006); severity of depressive symptoms was assessed using the Centre for Epidemiological Studies-Depression questionnaire (CES-D).’

Methods, Study variables, first paragraph, fourth sentence: The abbreviation ‘COPE’ needs to be defined at first mention.

Methods, Study variables, first paragraph, fifth sentence: The reference here, presumably WHO 2007, needs to be reviewed for style.

Methods, Study variables, fifth paragraph, first sentence: Recommend removing ‘International Index of Erectile Function’ in favour of abbreviation ‘IIEF’, as this has been previously defined.

Methods, Study variables, fifth paragraph, first sentence: Recommend removing ‘Female Sexual Functioning Index’ in favour of abbreviation ‘FSFI’, as this has been previously defined.

Methods, Study variables, fifth paragraph, first sentence: Recommend changing ‘…and determination of their internal consistency…’ to ‘…and subsequent determination of their internal consistency…’

Methods, Study variables, fifth paragraph, first sentence: Please change ‘Cronbach alpha’ to ‘Cronbach’s alpha’.

Methods, Study variables, fifth paragraph, second sentence: Please change ‘undertaken’ to ‘undertake’.

Methods, Study variables, fifth paragraph, third sentence: Please change ‘consistence’ to ‘consistency’.

Methods, Study variables, fifth paragraph, fifth sentence: Again, please remove the whole name of measures in favour of their abbreviations, IIEF and FSFI, since these have been previously defined.

Methods, Statistical analysis, first paragraph, fourth sentence: Please change to “sexual dysfunction for men” and “sexual dysfunction for women” as these are separate variables.

Methods, Statistical analysis, first paragraph, fifth sentence: Suggest changing to ‘To investigate the association of sexual dysfunction with socio-demographic, psychosocial and psychiatric illness factors…’

Methods, Statistical analysis, second paragraph: Suggest changing ‘at bivariate analyses’ to ‘with bivariate analyses’.

Results, Participant characteristics, second sentence: Suggest changing to ‘The majority of the participants…’

Table 1: ‘Moslems’ should be ‘Muslims’

Table 1: ‘ses’ should be ‘SES’ (three instances)

Results, Nature and prevalence of sexual dysfunction, second paragraph, second sentence: Suggest changing to ‘The majority of both men (89.3%) and women (66.3%) did not seek help for sexual dysfunction, but more women (25%) sought help from…’

Results, Nature and prevalence of sexual dysfunction, second paragraph, third sentence: Suggest changing ‘Of note 3.4% and 6.6%’ to ‘Of note, 3.4% and 6.6%’.

Table 2a: Please check the bolding of certain words and letters. This does not seem to be for emphasis or, if it is, has been applied incorrectly.

Table 2a, Sexual Pain problems, third item: ‘Over the past 4 weeks’ is unnecessary as it’s indicated at the top of the table.

Table 2b: Please check the underline of certain words. Is this necessary?

Table 2b, Sexual dysfunction, second row: Please delete empty row.

Results, Factors associated with sexual dysfunction, first paragraph, second sentence: Suggest changing to ‘A marital status indicated as widowed or separated…’

Results, Factors associated with sexual dysfunction, first paragraph, second sentence: Please change ‘Table 3’ to ‘Table 3 and 4’ since these results are the same using bivariate or multivariate analysis.

Results, Factors associated with sexual dysfunction, first paragraph, fourth sentence: Please change ‘Table 5’ to ‘Table 4’. There is no Table 5.

Discussion, general: Please replace ‘sexual dysfunction’ with ‘SD’ as the abbreviation is well-defined at this stage.

Discussion, first paragraph, first sentence: Suggest changing to ‘We sought to explore the topic of sexual functioning among patients living with HIV in Uganda, a rather taboo topic in this setting, with an aim of informing…’

Discussion, first paragraph, second and third sentences: Suggest changing to ‘SD is a relatively common problem experienced by men and women living with HIV in Uganda, with one in five men and two in five women affected. Studies on SD in PLWHA…’

Discussion, first paragraph, fourth sentence: Suggest changing to ‘It is worth noting that for both men and women with SD, the majority did not seek treatment. This was especially so for men, with less…’

Discussion, Nature and prevalence of dysfunction among PLWHA, second sentence: Please change ‘dysfunctions’ to ‘dysfunction’.

Discussion, Nature and prevalence of dysfunction among PLWHA, fourth sentence: Suggest changing to ‘This finding is similar to what has been reported in previous studies which found that the healthcare system may not be prepared for immediate or continuing support…’

Discussion, Factors associated with sexual dysfunction in HIV infected men, second paragraph, eighth sentence: Please change ‘2’ to ‘two’.

Discussion, Factors associated with sexual dysfunction in HIV infected men, second paragraph, ninth sentence: Please remove duplication of ‘on the’ in ‘on the on the other hand’.

Discussion, Factors associated with sexual dysfunction in HIV infected women, second paragraph, second sentence: Suggest changing to ‘In our study, a number of sexual problems, relating to reduced desire for and difficulties during sexual activity, were experienced by women. These findings are similar to findings…’

Discussion, Factors associated with sexual dysfunction in HIV infected women, second paragraph, fourth sentence: Suggest changing to ‘Concerns about HIV transmission, HIV treatment including ART and its side-effects…’

Discussion, Factors associated with sexual dysfunction in HIV infected women, second paragraph, fifth sentence: Suggest changing to ‘This could be a possible explanation why one in three women who participated…’

Discussion, Factors associated with sexual dysfunction in HIV infected women, third paragraph, first sentence: Suggest changing to ‘We found that older age (>45 years) and being widowed or separated…’

Discussion, Factors associated with sexual dysfunction in HIV infected women, third paragraph, second sentence: Suggest changing to ‘Oyedokun and others (2014) found no association between FSD and religion among HIV-positive women in Nigeria, however, increasing age (Oyedokun, Odeigah et al. 2014), was significantly associated with FSD (Luzi, Guaraldi et al. 2009). These findings are corroborated in our study.’

Discussion, Factors associated with sexual dysfunction in HIV infected women, third paragraph, third sentence: Please add a space after the bracket in ‘(Luzi, Guaraldi et al. 2009) and one can’.

Discussion, Factors associated with sexual dysfunction in HIV infected women, fifth paragraph, first sentence: Please add a space before the bracket in ‘higher (Wilson, Jean-Louis et al. 2010)’.

Discussion, Factors associated with sexual dysfunction in HIV infected women, sixth paragraph, first sentence: Suggest changing to ‘Psychosocial variables such as mental health status and quality of life, but not clinical status (illness stage), have been reported…’

References: References should be numbered and their style should be standardised throughout the manuscript.

Abstract

Sentence starting with ‘However, it is rarely discussed…’: Recommend replacing ‘secretive’ with ‘private’.

Introduction

Introduction, second paragraph, first sentence: Recommend changing to ‘Normal sexuality if a psycho-biological activity necessary for psychological and biological balance in most humans, and for the continuation of human life.’ It is dangerous to make blanket statements about sexuality being necessary for all people as it marginalises those who cannot, or choose not to, engage with their sexuality.

Introduction, third paragraph, section from ‘PLWHA are living longer and healthier lives…’ up to ‘…unintended pregnancy (Harrison and O’Sullivan 2010, Okigbo and Speizer 2015, Wasswa, Kabagenyi et al. 2020). In the first instance, the references should be standardised to a single format. In the second instance, this seems to be a repetition and expansion of what was said in the first paragraph of the introduction. Recommend working in the contents of the first paragraph into this one, resulting in the paragraph starting with ‘Normal sexuality…’ being the first of the Introduction.

Methods

Methods, Study design, second sentence: Please explain why sexual is written in inverted commas (‘sexual’ relationships). I have not been able to find a rationale for this in the manuscript. It seems unnecessary and could alienate readers.

Methods, Statistical analysis, general: I can appreciate that this may have been difficult to ascertain and potentially impossible to obtain reliable reporting on, given the setting, but did the authors consider subgroup effects by homo- and heterosexual respondents? While I understand potential legal ramifications of exploring this, it should at least be noted in the limitations as important subgroup effects might exist – particularly since major depressive disorder is a risk factor for sexual dysfunction. It is likely that MDD disproportionately affects homosexual people, particularly in a context where they are not free to express their sexuality.

Results

Results, Participant characteristics: Please make some mention of the disproportionate representation of women in the sample. Conclusions based on a sample of 159 men, particularly when juxtaposed or compared with women, could be spurious, and should be noted.

Results, Nature and prevalence of sexual dysfunction, first paragraph, third sentence: ‘number of times sexual intercourse was attempted’ is listed as a prevalent problem. What does this mean? Is a high number of attempts bad or indicative or sexual dysfunction?

Table 3: It is not clear from this table or the accompanying methodological text why certain parameters are set at 1 with no 95% confidence interval indicated. Were these variables set as such in the model? This needs to be explained.

Table 3: It is not clear why no results of HIV neurocognitive impairment is given for men. This needs to be described, either by a footnote to the table or in text.

Table 4: Please provide the results of stroke from the bivariate model in Table 3, as it is not listed there. Stroke, presumably previous stroke, shows significant association with SD for both men and women, and the results from the bivariate analysis should be shown. In addition, the outcome should be capitalised to ‘Stroke’ and the significance level for men should be shown as ‘P=0.023’.

Table 4: Bivariate analysis of diabetes in men meets the a priori cut-off of P=0.2 for inclusion in the multivariate model. Why are these results not presented?

Results, Factors associated with sexual dysfunction: It is not clear why some results from the bivariate model and some results from the multivariate model are reported. Results from the multivariate model, representing the most adjusted measures, should be used preferentially – unless there is a reason not to do so. If the latter is the case, this needs to be explicitly explained. If both results need to be reported, they should be clearly identified according to the model it is derived from.

Results, Factors associated with sexual dysfunction, first paragraph: In addition to other socio-demographic variables listed, employment is also significantly associated with SD in women at the 5% level.

Results, Factors associated with sexual dysfunction, first paragraph: It is not clear why only some numeric results from statistical models are reported in text.

Results, Factors associated with sexual dysfunction, first paragraph, third sentence: Please change ‘0.61’ to ‘1.61’ in accordance with the result in Table 4.

Results, Factors associated with sexual dysfunction, second paragraph, first sentence: Suggest changing to ‘With respect to men, religion (being of the Muslim or another faith) was the only…’

Results, Factors associated with sexual dysfunction, second paragraph: In addition to other socio-demographic variables listed, age is also highly significantly associated with SD in men. Why is this not mentioned here?

Results, Factors associated with sexual dysfunction, second paragraph: Stroke is also significantly associated with SD in men according to the results presented in Table 4. This needs to be reported here, particularly in the light of the role of the vascular system in erectile function.

Results, Factors associated with sexual dysfunction, third paragraph: It is not clear why this information is presented here. The text relating to the ages of respondents should be placed at the start of the Results section, though the age of male respondents should also be reported. Furthermore, the categorisation of education and religion should be described in the Methods section. Finally, it is not clear why this section describes the categorisation of religion as ‘Christian and non-Christian’ when all other sections report ‘Christian’ and ‘Muslim and others’.

Discussion

Discussion, Factors associated with sexual dysfunction in HIV infected men, general: The authors state that their study did not find associations with physical or anthropometric factors and SD. However, both stroke (in the multivariate model) and diabetes (in the bivariate model, with no results for multivariate analysis despite meeting the a priori cut-off) are significantly associated with male SD. Both of these also have known vascular causes and/or effects; the vascular system, in turn, is arguably the most important system for erectile function. As such, the statement that no associations between physical factors and SD were found is patently incorrect.

Discussion, Factors associated with sexual dysfunction in HIV infected men, second paragraph, sixth sentence: It is not clear what the (10) at the end of this sentence refers to. The next sentence seems to follow on from this, stating ‘In that study’, but reference 10 in the manuscript does not correspond to Lallemand, Salhi et al. 2002.

Discussion, Factors associated with sexual dysfunction in HIV infected men, second paragraph, eighth sentence: The authors state the no association was found with ART, however, ‘ART’ as an explanatory variable was not measured – only ART adherence. Please specify here whether ART adherence is the intention, or whether some other dimension of ‘ART’ was measured and its association tested.

Discussion, Factors associated with sexual dysfunction in HIV infected women, third paragraph, general: It is noticeable how much discussion is had around the causal pathways of age in SD in women living with HIV, yet nothing is said about the effect of age for men. This is a glaring omission.

Limitations

As noted previously, it is acknowledged that the legal environment in Uganda may have precluded the accurate gathering of information around homo- and heterosexuality among the sample. However, there may be important subgroup effects and this possibility should be noted here.

Author contributions

The authors state that detailed contributions are to be listed under the section in the second sentence, however, no such information is forthcoming. The only explicit contribution that is listed is analysis and interpretation by WS. This is not sufficient detail to determine whether ICMJE authorship criteria have been met.

6. PLOS authors have the option to publish the peer review history of their article (what does this mean?). If published, this will include your full peer review and any attached files.

Reviewer #1: No

---

## [Author Response · Author response to Decision Letter 0]

28 Dec 2022

Dear Sir/Madam,

Response to review comments

Comment Response

General

Abstract, sentence starting with ‘Among individuals living with HIV/AIDS…’: Suggest adding changing

to ‘…have hardly been studied, especially in sub-Saharan Africa…’ Thank you for the comment. The change has been effected as advised. Page 1 

[Lack of] hyphenation is a recurring problem in the manuscript:

• Recommend changing first sentence of Abstract to read ‘We conducted a clinic-based cross-sectional

survey among 710 PLWHA in stable ‘sexual’ relationships in central and southwestern

Uganda. The first sentence of abstract has been changed to read as: We conducted a clinic-based cross-sectional survey among 710 people living with HIV/AIDS in stable ‘sexual’ relationships in central and southwestern Uganda.

• Recommend changing ‘HIV related’ to ‘HIV-related’ in the Abstract, Methods section and

throughout the manuscript ‘HIV related’ has been changed to ‘HIV-related’ throughout the manuscript

• Please change ‘HIV infected’ to ‘HIV-infected’ throughout the manuscript. ‘HIV infected’ has been changed to ‘HIV-infected’ throughout the manuscript

• Please change ‘HIV positive’ to ‘HIV-positive’ throughout the manuscript. ‘HIV positive’ has been changed to ‘HIV-positive’ throughout the manuscript

Abstract, Methods section: Suggest changing to ‘…psychosocial factors (maladaptive coping styles,

negative life events, social support, resilience, HIV stigma)…’ The change has been effected as advised Page 1

Inappropriate capitalisation is a recurring problem in the manuscript: Recommend changing sentence in Abstract, Methods section to ‘…and clinical factors (CD4 counts, body weight, height, HIV clinical stage, treatment adherence).’ The sentence has been changed as recommended

Recommend changing to ‘negative life events’, ‘social support’, and ‘resilience’ in Methods,

Study variables, fourth sentence. The sentence has been changed as recommended

Introduction, third paragraph, fifth sentence: Categories of sexual dysfunction do not need to be capitalised. Thank you for the observation. The capitalisation has been removed

Methods, Study variables, fifth sentence: Recommend changing to ‘body weight’ and ‘height’. Thank you. The change has been effected as recommended 

Table 1: ‘Others’ in ‘Moslems & Others’ don’t need to be capitalised. The capitalisation has been removed

Table 2a, Orgasm problems, second item: ‘How’ does not need to be capitalised. The capitalisation has been removed

Results, Factors associated with sexual dysfunction, first paragraph, third sentence: ‘Depressive’ in ‘Major Depressive disorder’ does not need to be capitalised. The capitalisation has been removed

Abstract, first sentence of Results: Suggest changing to ‘SD was more prevalent in women (38.7%) than men (17.6%) and the majority (89.3% of men and 66.3% of women)…’ The change has been effected as suggested 

Abstract, first sentence of Conclusion: Suggest changing to ‘Sexual dysfunction has considerable prevalence among PLWHA in Uganda. It is associated with socio-demographic, psychiatric and clinical illness factors. To further improve the quality of life of PLWHA, they should be screened for sexual dysfunction as part of routine assessment.’ The change has been effected as recommended

Introduction, first paragraph, second sentence: Suggest changing to ‘As a result, attention has now shifted to…’ The change has been effected as recommended

Introduction, first paragraph, fourth sentence: Suggest changing to ‘Sexual function in the context of HIV/AIDS has mainly been studied with respect to…’ The change has been effected as recommended

Introduction, first paragraph, fifth sentence: Suggest changing to ‘From a public health perspective, sexual problems may predispose individuals to unsafe sexual practices as well as intimate partner relationship discord, including marital violence, and poor ART adherence. The change has been effected as recommended

Introduction, first paragraph, seventh sentence: Recommend changing ‘ill mental health’ to ‘poor mental health’. The change has been effected as recommended

Introduction, second paragraph, second sentence: Suggest changing to ‘It involves the feelings of desire, behavior that brings pleasure to oneself and one’s partner, and stimulation of the primary sex organs, including coitus. The activity should be devoid of…’ The change has been effected as recommended

Introduction, second paragraph, third sentence: Suggest changing to ‘Anatomy, physiology, psychology, the culture in which one lives, relationships with others, and the developmental

experiences throughout the life cycle determines sexuality.’

Introduction, third paragraph, third sentence: Please replace ‘FSD’ with ‘SD’. The change has been effected as recommended

Methods, Study setting, second paragraph, third sentence: Suggest changing to ‘On the other hand,

TASO Entebbe commenced its work in 1991 and…’ The change has been effected as recommended

Methods, Study setting, second paragraph, fourth sentence: Suggest changing to ‘Entebbe, being urbanized, has an ever-growing and transient population including traders, fish-mongers, and uniformed personnel. It is also a first and last stop for international tourists.’ The change has been effected as recommended

Methods, Study variables, first paragraph, first sentence: Suggest changing to ‘A range of explanatory variables that included socio-demographic, psychiatric illness, and clinical factors were assessed.’ The change has been effected as recommended

Methods, Study variables, first paragraph, second sentence: Suggest changing ‘…constructed from common household items…’ to ‘…constructed from having common household items…’ The change has been effected as recommended

Methods, Study variables, first paragraph, third sentence: Suggest changing to ‘The psychiatric illness factors included major depressive disorder and suicidality (assessed using the Mini International Neuropsychiatric interview (M.I.N.I. Plus) (Sheehan and Lecrubier 2006); severity of depressive symptoms was assessed using the Centre for Epidemiological Studies-Depression questionnaire (CESD).’ The change has been effected as recommended

Methods, Study variables, first paragraph, fourth sentence: The abbreviation ‘COPE’ needs to be defined at first mention. The abbreviation COPE has been defined as ‘Coping Orientation to Problems Experienced’

Methods, Study variables, first paragraph, fifth sentence: The reference here, presumably WHO 2007, needs to be reviewed for style. The reference has been revised to WHO 2007

Methods, Study variables, fifth paragraph, first sentence: Recommend removing ‘International Index of Erectile Function’ in favour of abbreviation ‘IIEF’, as this has been previously defined. The change has been effected as recommended

Methods, Study variables, fifth paragraph, first sentence: Recommend removing ‘Female Sexual Functioning Index’ in favour of abbreviation ‘FSFI’, as this has been previously defined. The change has been effected as recommended

Methods, Study variables, fifth paragraph, first sentence: Recommend changing ‘…and determination of their internal consistency…’ to ‘…and subsequent determination of their internal consistency…’ The change has been effected as recommended

Methods, Study variables, fifth paragraph, first sentence: Please change ‘Cronbach alpha’ to ‘Cronbach’s alpha’. The change has been effected as recommended

Methods, Study variables, fifth paragraph, second sentence: Please change ‘undertaken’ to ‘undertake’. The change has been effected as recommended

Methods, Study variables, fifth paragraph, third sentence: Please change ‘consistence’ to ‘consistency’. The change has been effected as recommended

Methods, Study variables, fifth paragraph, fifth sentence: Again, please remove the whole name of measures in favour of their abbreviations, IIEF and FSFI, since these have been previously defined. The change has been effected as recommended

Methods, Statistical analysis, first paragraph, fourth sentence: Please change to “sexual dysfunction for men” and “sexual dysfunction for women” as these are separate variables. The change has been effected as recommended

Methods, Statistical analysis, first paragraph, fifth sentence: Suggest changing to ‘To investigate the association of sexual dysfunction with socio-demographic, psychosocial and psychiatric illness factors…’ The change has been effected as recommended

Methods, Statistical analysis, second paragraph: Suggest changing ‘at bivariate analyses’ to ‘with bivariate analyses’. The change has been effected as recommended

Results, Participant characteristics, second sentence: Suggest changing to ‘The majority of the participants…’ The change has been effected as recommended

Table 1: ‘Moslems’ should be ‘Muslims’ The change has been effected as recommended

Table 1: ‘ses’ should be ‘SES’ (three instances) The change has been effected as recommended

Results, Nature and prevalence of sexual dysfunction, second paragraph, second sentence: Suggest changing to ‘The majority of both men (89.3%) and women (66.3%) did not seek help for sexual dysfunction, but more women (25%) sought help from…’ The change has been effected as recommended

Results, Nature and prevalence of sexual dysfunction, second paragraph, third sentence: Suggest changing ‘Of note 3.4% and 6.6%’ to ‘Of note, 3.4% and 6.6%’. The change has been effected as recommended

Table 2a: Please check the bolding of certain words and letters. This does not seem to be for emphasis or, if it is, has been applied incorrectly. The bolding has been removed

Table 2a, Sexual Pain problems, third item: ‘Over the past 4 weeks’ is unnecessary as it’s indicated at the top of the table. It has been removed

Table 2b: Please check the underline of certain words. Is this necessary? The underline has been removed

Table 2b, Sexual dysfunction, second row: Please delete empty row. The empty row has been deleted

Results, Factors associated with sexual dysfunction, first paragraph, second sentence: Suggest changing to ‘A marital status indicated as widowed or separated…’ The change has been effected as recommended

Results, Factors associated with sexual dysfunction, first paragraph, second sentence: Please changes ‘Table 3’ to ‘Table 3 and 4’ since these results are the same using bivariate or multivariate analysis. ‘Table 3’ has been changed to ‘Table 3 and 4’

Results, Factors associated with sexual dysfunction, first paragraph, fourth sentence: Please change ‘Table 5’ to ‘Table 4’. There is no Table 5. ‘Table 5’ has been changed to ‘Table 4’

Discussion, general: Please replace ‘sexual dysfunction’ with ‘SD’ as the abbreviation is well-defined at this stage. The change has been effected as recommended

Discussion, first paragraph, first sentence: Suggest changing to ‘We sought to explore the topic of sexual functioning among patients living with HIV in Uganda, a rather taboo topic in this setting, with an aim of informing…’ The change has been effected as recommended

Discussion, first paragraph, second and third sentences: Suggest changing to ‘SD is a relatively common problem experienced by men and women living with HIV in Uganda, with one in five men and two in five women affected. Studies on SD in PLWHA…’ The change has been effected as recommended

Discussion, first paragraph, fourth sentence: Suggest changing to ‘It is worth noting that for both men and women with SD, the majority did not seek treatment. This was especially so for men, with less…’ The change has been effected as recommended

Discussion, Nature and prevalence of dysfunction among PLWHA, second sentence: Please change ‘dysfunctions’ to ‘dysfunction’. The change has been effected as recommended

Discussion, Nature and prevalence of dysfunction among PLWHA, fourth sentence: Suggest changing to ‘This finding is similar to what has been reported in previous studies which found that the healthcare system may not be prepared for immediate or continuing support…’ The change has been effected as recommended

Discussion, Factors associated with sexual dysfunction in HIV infected men, second paragraph, and eighth sentence: Please change ‘2’ to ‘two’. The change has been effected as recommended

Discussion, Factors associated with sexual dysfunction in HIV infected men, second paragraph, ninth Sentence: Please remove duplication of ‘on the’ in ‘on the on the other hand’. The change has been effected as recommended

Discussion, Factors associated with sexual dysfunction in HIV infected women, second paragraph, second sentence: Suggest changing to ‘In our study, a number of sexual problems, relating to reduced desire for and difficulties during sexual activity, were experienced by women. These findings are similar to findings…’ The change has been effected as recommended

Discussion, Factors associated with sexual dysfunction in HIV infected women, second paragraph, fourth sentence: Suggest changing to ‘Concerns about HIV transmission, HIV treatment including ART and its side-effects…’ The change has been effected as recommended

Discussion, Factors associated with sexual dysfunction in HIV infected women, second paragraph, and fifth sentence: Suggest changing to ‘This could be a possible explanation why one in three women who participated…’ The change has been effected as recommended

Discussion, Factors associated with sexual dysfunction in HIV infected women, third paragraph, and first sentence: Suggest changing to ‘We found that older age (>45 years) and being widowed or separated…’ The change has been effected as recommended

Discussion, Factors associated with sexual dysfunction in HIV infected women, third paragraph, second sentence: Suggest changing to ‘Oyedokun and others (2014) found no association between FSD and religion among HIV-positive women in Nigeria, however, increasing age (Oyedokun, Odeigah et al. 2014), was significantly associated with FSD (Luzi, Guaraldi et al. 2009). These findings are corroborated in our study.’ The change has been effected as recommended

Discussion, Factors associated with sexual dysfunction in HIV infected women, third paragraph, third sentence: Please add a space after the bracket in ‘(Luzi, Guaraldi et al. 2009) and one can’. A space has been added

Discussion, Factors associated with sexual dysfunction in HIV infected women, fifth paragraph, first sentence: Please add a space before the bracket in ‘higher (Wilson, Jean-Louis et al. 2010)’. The change has been effected as recommended

Discussion, Factors associated with sexual dysfunction in HIV infected women, sixth paragraph, first sentence: Suggest changing to ‘Psychosocial variables such as mental health status and quality of life, but not clinical status (illness stage), have been reported…’ The change has been effected as recommended

References: References should be numbered and their style should be standardised throughout the manuscript. The references have been numbered

Abstract 

Sentence starting with ‘However, it is rarely discussed…’: Recommend replacing ‘secretive’ with ‘private’. This seems to be a more appropriate word as sexual life is considered private by many, but not necessarily a secret. The change has been effected as recommended

Introduction 

Introduction, second paragraph, first sentence: Recommend changing to ‘Normal sexuality if a psychobiological activity necessary for psychological and biological balance in most humans, and for the continuation of human life.’ It is dangerous to make blanket statements about sexuality being necessary for all people as it marginalises those who cannot, or choose not to, engage with their sexuality. The change has been effected as recommended

Introduction, third paragraph, section from ‘PLWHA are living longer and healthier lives…’ up to‘…unintended pregnancy (Harrison and O’Sullivan 2010, Okigbo and Speizer 2015, Wasswa, Kabagenyi et al. 2020). In the first instance, the references should be standardised to a single format. In the second instance, this seems to be a repetition and expansion of what was said in the first paragraph of the introduction. Recommend working in the contents of the first paragraph into this one, resulting in the paragraph starting with ‘Normal sexuality…’ being the first of the Introduction. The section has been revised and the sentence on normal sexuality is now the first one in the introduction

Methods 

Methods, Study design, the second sentence: Please explain why sexual is written in inverted commas (‘sexual’ relationships). I have not been able to find a rationale for this in the manuscript. It seems unnecessary and could alienate readers. Thank you for the comment. The inverted commas have been removed. 

Methods, Statistical analysis, general: I can appreciate that this may have been difficult to ascertain and potentially impossible to obtain reliable reporting on, given the setting, but did the authors consider subgroup effects by homo- and heterosexual respondents? While I understand potential legal ramifications of exploring this, it should at least be noted in the limitations as important subgroup effects might exist – particularly since major depressive disorder is a risk factor for sexual dysfunction. It is likely that MDD disproportionately affects homosexual people, particularly in a context where they are not free to express their sexuality. We were not able to conduct subgroup analyses by homo- and heterosexual respondents. In Uganda, homosexuality is still a sensitive issue to discuss.

Results 

Results, Participant characteristics: Please make some mention of the disproportionate representation of women in the sample. Conclusions based on a sample of 159 men, particularly when juxtaposed or compared with women, could be spurious, and should be noted. A sentence has been added to the 3rd paragraph in the discussion section ‘However, the number of men was smaller than that of women in this study. This could have affected the results.’

Results, Nature and prevalence of sexual dysfunction, first paragraph, third sentence: ‘number of times sexual intercourse was attempted’ is listed as a prevalent problem. What does this mean? Is a high number of attempts bad or indicative or sexual dysfunction? The number of times sexual intercourse was attempted was measured as” no attempts, one to two attempts/ three to four attempts, five to six attempts, seven to ten attempts/ Eleven or more attempts”. In this case, fewer sexual attempts were assumed to indicate lower sex drive and, therefore sexual dysfunction.

Table 3: It is not clear from this table or the accompanying methodological text why certain parameters are set at 1 with no 95% confidence interval indicated. Were these variables set as such in the model? This needs to be explained. The parameters that were set at 1 are the comparison factors. The odds of these other parameters/categories are interpreted in relation to the comparison parameters

Table 3: It is not clear why no results of HIV neurocognitive impairment are given for men. This needs to be described, either by a footnote to the table or in text. * HIV neurocognitive impairment among men was not included due to perfect collinearity. This has been added as a footnote

Table 4: Please provide the results of stroke from the bivariate model in Table 3, as it is not listed there. Stroke, presumably previous stroke, shows significant association with SD for both men and women, and the results from the bivariate analysis should be shown. In addition, the outcome should be capitalised to ‘Stroke’ and the significance level for men should be shown as ‘P=0.023’. Thank you for the observation. The results have been included in table 3

Table 4: Bivariate analysis of diabetes in men meets the a priori cut-off of P=0.2 for inclusion in the multivariate model. Why are these results not presented? Some of the factors met the apriori cut-off. However they presented multicollinearity in the multivariate model and therefore reducing precision of the estimates and were therefore eliminated.

Results, Factors associated with sexual dysfunction: It is not clear why some results from the bivariate model and some results from the multivariate model are reported. Results from the multivariate model, representing the most adjusted measures, should be used preferentially – unless there is a reason not to do so. If the latter is the case, this needs to be explicitly explained. If both results need to be reported, they should be clearly identified according to the model it is derived from. Some of the factors met the apriori cut-off. However they presented multicollinearity in the multivariate model and therefore reducing precision of the estimates and were therefore eliminated.

Results, Factors associated with sexual dysfunction, first paragraph: In addition to other sociodemographic variables listed, employment is also significantly associated with SD in women at the 5% level. This has been indicated

Results, Factors associated with sexual dysfunction, first paragraph: It is not clear why only some numeric results from statistical models are reported in text. The details of the results are in the table. It would have been repetition to report all results in text and table

Results, Factors associated with sexual dysfunction, first paragraph, third sentence: Please change ‘0.61’ to ‘1.61’ in accordance with the result in Table 4. Thank you for the observation. This has been addressed

Results, Factors associated with sexual dysfunction, second paragraph, first sentence: Suggest changing to ‘With respect to men, religion (being of the Muslim or another faith) was the only…’ This has been changed. Thanks for the comment. This has been addressed

Results, Factors associated with sexual dysfunction, second paragraph: In addition to other sociodemographic variables listed, age is also highly significantly associated with SD in men. Why is this not mentioned here? Age has been included. Thank you for the observation

Results, Factors associated with sexual dysfunction, second paragraph: Stroke is also significantly associated with SD in men according to the results presented in Table 4. This needs to be reported here, particularly in the light of the role of the vascular system in erectile function. It has been included. Thank you

Results, Factors associated with sexual dysfunction, third paragraph: It is not clear why this information is presented here. The text relating to the ages of respondents should be placed at the start of the Results section, though the age of male respondents should also be reported The information has been removed from this section and taken to methods and the beginning of the results section

Furthermore, the categorisation of education and religion should be described in the Methods section. This has been taken to the end of the methods section

Finally, it is not clear why this section describes the categorisation of religion as ‘Christian and non-Christian’ when all other sections report ‘Christian’ and ‘Muslim and others’. This has been harmonised. Religion is categorised as Christian and, muslim and other religion

Discussion 

Discussion, Factors associated with sexual dysfunction in HIV infected men, general: The authors state that their study did not find associations with physical or anthropometric factors and SD. However, both stroke (in the multivariate model) and diabetes (in the bivariate model, with no results for multivariate analysis despite meeting the a priori cut-off) are significantly associated with male SD. Both of these also have known vascular causes and/or effects; the vascular system, in turn, is arguably the most important system for erectile function. As such, the statement that no associations between physical factors and SD were found is patently incorrect. Thank you so much for the observarion . it was a wrong statement and it has been corrected. Indeed, diabetes and stroke and known risk factors for sexual dysfunction. In our results, they were associated with SD.

Discussion, Factors associated with sexual dysfunction in HIV infected men, second paragraph, sixth sentence: It is not clear what the (10) at the end of this sentence refers to. The next sentence seems to follow on from this, stating ‘In that study’, but reference 10 in the manuscript does not correspond to Lallemand, Salhi et al. 2002. The mistake has been corrected. The reference is Lallemand et al 2002

Discussion, Factors associated with sexual dysfunction in HIV infected men, second paragraph, eighth sentence: The authors state the no association was found with ART, however, ‘ART’ as an explanatory

variable was not measured – only ART adherence. Please specify here whether ART adherence is the intention, or whether some other dimension of ‘ART’ was measured and its association tested. Thank you for the comment. The section refers to ART adherence and has been edited

Discussion, Factors associated with sexual dysfunction in HIV infected women, third paragraph, general: It is noticeable how much discussion is had around the causal pathways of age in SD in women living with HIV, yet nothing is said about the effect of age for men. This is a glaring omission. Thank you for this comment. A section has been added on the effect of age on sexual dysfunction among men.

Limitations 

General: As noted previously, it is acknowledged that the legal environment in Uganda may have precluded the accurate gathering of information around homo- and heterosexuality among the sample. However, there may be important subgroup effects and this possibility should be noted here. Thank you for the comment. We did not do subgroup analysis. It is possible that there could have been important subgroup effects

Author contributions 

General: The authors state that detailed contributions are to be listed under the section in the second sentence, however, no such information is forthcoming. The only explicit contribution that is listed is analysis and interpretation by WS. This is not sufficient detail to determine whether ICMJE authorship criteria have been met. Thank you for this reminder. It has been updated

---

## [Decision Letter · Decision Letter 1]

22 May 2023

PONE-D-22-24725R1A ‘hidden problem’: Nature, prevalence and factors associated with sexual dysfunction in persons living with HIV/AIDS in UgandaPLOS ONE

Dear Dr. Godfrey Zari Rukundo,

Thank you for submitting your manuscript to PLOS ONE. After careful consideration, we feel that it has merit but does not fully meet PLOS ONE’s publication criteria as it currently stands. Therefore, we invite you to submit a revised version of the manuscript that addresses the points raised during the review process.

ACADEMIC EDITOR: Your work requires thorough editorial revision from grammar to in-text citations to referencing style.Revise the Methods and Discussion sections

We look forward to receiving your revised manuscript.

Kind regards,

Habil Otanga, Ph.D

Academic Editor

PLOS ONE

Additional Editor Comments (if provided):

1. Your work suffers serious editorial shortcomings, from grammar to in-text citations to referencing style.

2. Methods: ED was measured using 6 items on the IIEF. How was the cut-off score of 25 derived (on a 5 point Likert Scale) which is erroneously presented as a 6 point Likert Scale (Section on Study Variables Line 27)? What informed the cut-off score? Reviewers found fault with sampling techniques and determination of sample size, data collection tools and techniques.

Why was the definition for SD different for men and women judging by the fact that authors used a single subscale for men (6 items) and the total FSFI score for women?

3. Discussion/Results: All tables must be presented in APA format

-If scales were used, one would expect Table 2 (a, b) to present frequencies/means of variables and not items. Why were the tables presented item by item? When do variables cease to be items and become variables for analysis? What was used in analysis - items in the questionnaire or variables? Reviewers suggest that authors discuss the statistical part of how categorical and continuous data are reported

Reviewers' comments:

Reviewer's Responses to Questions

**Comments to the Author**

1. If the authors have adequately addressed your comments raised in a previous round of review and you feel that this manuscript is now acceptable for publication, you may indicate that here to bypass the “Comments to the Author” section, enter your conflict of interest statement in the “Confidential to Editor” section, and submit your "Accept" recommendation.

Reviewer #2: (No Response)

Reviewer #3: (No Response)

2. Is the manuscript technically sound, and do the data support the conclusions?

Reviewer #2: Partly

Reviewer #3: Yes

3. Has the statistical analysis been performed appropriately and rigorously? 

Reviewer #2: No

Reviewer #3: Yes

4. Have the authors made all data underlying the findings in their manuscript fully available?

Reviewer #2: Yes

Reviewer #3: Yes

5. Is the manuscript presented in an intelligible fashion and written in standard English?

Reviewer #2: Yes

Reviewer #3: Yes

6. Review Comments to the Author

Reviewer #2: Thanks for your paper. Whereas the paper presents an amazing perspective to sexual dysfunction among persons living with HIV in Uganda, there are issues that you need to address:

1. Editing: Your entire work suffers from poor editorial work. Words that are not proper nouns start with capital letters e.g., Line 2 of abstract.

-Line 2 of abstract is either incomplete or poorly punctuated (Sentence beginning with "Although"

2. Citations: You need to revise the entire work to be in line with the appropriate journal citation format.

Introduction: Line 5

Paragraph 2 of Introduction: Line 1, 3, 7, 9

Paragraph 3: Line 5, 8, 10, 12. for instance what citation style is "Wilson, Jean-Louis et al 2010"? Decide who the first author is, use their surname followed by et al and year of publication.

3. Methods: ED was measured using 6 items on the IIEF. How was the cut-off score of 25 derived (on a 5 point LS) which is erroneously presented as a 6 point LS (Section on Study Variables Line 27)? What informed the cut-off score?

Why was the definition for SD different for men and women judging by the fact that using a single subscale for men (6 items) and the total FSFI score for women?

4. Results: All tables must be presented in APA format

-If scales were used, one would expect Table 2 (a, b) to present frequencies/means of variables and not items. Why were the tables presented item by item? When do variables cease to be items and become variables for analysis? What was used in analysis - items in the questionnaire or variables?

5. References do not follow any distinct style. Check PLoS ONE requirements for presenting references.

Reviewer #3: Dear Editor, I would like to thank you for inviting me to review this cross-sectional study titled "Nature, prevalence, and factors associated with sexual dysfunction in persons living with HIV/AIDS in Uganda." This study addresses a novel and little-researched issue that affects both the general public and people living with HIV/AIDS. Despite the fact that there were interesting concepts scattered throughout the work, I've highlighted a few of my complaints below with subheadings that the authors should take into account before considering publishing.

Abstract:

- By taking into account the study period, sampling technique, data collection tools, and statistical methodologies (descriptive or inferential statistics), the authors needed to update the methods part of the abstract once more in light of their objectives.

- The authors must indicate the percentage with a 95% confidence interval for the prevalence of sexual disfunction for males and females, respectively.

- In light of their findings, the authors ought to revise their recommendations.

Introduction:

- The introduction section focuses primarily on the definition, nature, and determinants of sexual dysfunction, with little to no discussion of its prevalence. I should suggest that the authors review the global literature on the prevalence in order to briefly explain the epidemiology (the distribution) of sexual dysfunction.

Methods:

-When was this study done? considering that the information came from a secondary source and that the patients with HIV/AIDS were chosen from medical records from what years?

- In addition to mentioning that this research was conducted as part of Prof Eugene Kinyanda’s Senior EDCTP Fellowship funded study entitled, ‘Clinical trials in HIV/AIDS in Africa: Should they routinely control for mental health factors?’, I believe the authors missed an important section of the materials and methods section, such as how the participants for the study were selected (sampling technique) and the determination of sample size, data collection tools and techniques, the operational definitions of some concepts, etc.

- The authors should thoroughly discuss the statistical part of how categorical and continuous data are reported, respectively, and for variables in the final multivariate model, the authors should plan the adjusted odds ratio (AOR) along with the 95% confidence interval. When measuring the outcome variable, the authors needed to clearly categorize it into binary data by coding it.

- The authors had to identify at what level of significance the variables should be deemed significant in the final multivariate model, which also needed to be assessed and presented together with an AOR and a 95% confidence interval.

- Additionally, the authors had to disclose the precise categorization (their levels) of each independent variable measured in addition to stating the scales.

Results:

- I'm not sure why the authors chose to use the median rather than the mean when reporting the age.

- The authors had to specify what low and high socioeconomic status meant when classifying socioeconomic status based on the wealth index in the method section before reporting here.

- It is required to give the mean, plus or minus standard deviation, or the sub-total frequency of each domain of measurement scales when reporting sexual dysfunction in males and females. Additionally, for each outcome variable, the authors needed to indicate the 95% confidence interval.

- I'll advise you to place the subheading factors related to sexual dysfunction before presenting the tables labelled "bivariate analysis of related factors (not adjusted) and "multivariate model (tables 3 and 4).

Discussion:

- The finding was well discussed by the authors. The authors were also required to expand on their discussion by including earlier prevalence research related to sexual dysfunction as well as further implications of the study for low-income nations like those in sub-Saharan Africa.

- Although it appears that the authors are talking about qualitative findings in their discussion, I believe that they should concentrate on quantitative results instead because they evaluated the study in a quantitative manner, unless they felt the need to include qualitative data.

Conclusion and recommendation:

- The authors should have provided a more thorough summary of the main findings and recommendations based on those findings, particularly for those stakeholders working on behavioural changes related to HIV/AIDS and for concerned policymakers.

At last, the authors produced some excellent work, but they should pay particular attention to the methods section because the majority of its components are missing.

7. PLOS authors have the option to publish the peer review history of their article (what does this mean?). If published, this will include your full peer review and any attached files.

Reviewer #2: No

Reviewer #3: No

---

## [Author Response · Author response to Decision Letter 1]

1 Nov 2023

October 27, 2023

Dr. Habil Otanga, Ph.D

Academic Editor

PLOS ONE 

Dear Dr. Otanga,

Re: PONE-D-22-24725R1: A ‘hidden problem’: Nature, prevalence and factors associated with sexual dysfunction in persons living with HIV/AIDS in Uganda

Thank you for the opportunity to resubmit this manuscript. Your effort and that of the reviewers is highly appreciated. Here below is a point-by-point response to the review comments.

Comment Response Page with changes

Editor’s comments

1. Your work suffers serious editorial shortcomings, from grammar to in-text citations to referencing style.

 Thak you for your comment. The manuscript has been revised and improved. Throughout the manuscript

2. Methods: ED was measured using 6 items on the IIEF. How was the cut-off score of 25 derived (on a 5-point Likert Scale) which is erroneously presented as a 6-point Likert Scale (Section on Study Variables Line 27)? What informed the cut-off score? Reviewers found fault with sampling techniques and determination of sample size, data collection tools and techniques.

Why was the definition for SD different for men and women judging by the fact that authors used a single subscale for men (6 items) and the total FSFI score for women?

 This has been corrected to a 15 item tool.

the cut off score was informed by a previous study (IIEF (Rosen, Riley et al. 1997))

The definition from ISD is different as assessed differently by the different tools i.e IIEF & FSFI Page 7

3. Discussion/Results: All tables must be presented in APA format

-If scales were used, one would expect Table 2 (a, b) to present frequencies/means of variables and not items. Why were the tables presented item by item? The APA format has been implemented Pages 11, 12,13,14,15,16,17,18

When do variables cease to be items and become variables for analysis? What was used in analysis - items in the questionnaire or variables? 

 Items from the questionnaire were used to generate composite variables for the outcomes.

Standard variables like sociodemographic were also used. 

Reviewers suggest that authors discuss the statistical part of how categorical and continuous data are reported This has been added Page 10

Reviewer #2: Thanks for your paper. Whereas the paper presents an amazing perspective to sexual dysfunction among persons living with HIV in Uganda, there are issues that you need to address:

1. Editing: Your entire work suffers from poor editorial work. Words that are not proper nouns start with capital letters e.g., Line 2 of abstract.

-Line 2 of abstract is either incomplete or poorly punctuated (Sentence beginning with "Although"

 Thank you for the observation. This has been addressed in the revised manuscript 

2. Citations: You need to revise the entire work to be in line with the appropriate journal citation format.

Introduction: Line 5

Paragraph 2 of Introduction: Line 1, 3, 7, 9

Paragraph 3: Line 5, 8, 10, 12. for instance what citation style is "Wilson, Jean-Louis et al 2010"? Decide who the first author is, use their surname followed by et al and year of publication.

 Thank you for the guidance. The citations have been revised as advised. 

3. Methods: ED was measured using 6 items on the IIEF. How was the cut-off score of 25 derived (on a 5 point LS) which is erroneously presented as a 6 point LS (Section on Study Variables Line 27)? What informed the cut-off score?

Why was the definition for SD different for men and women judging by the fact that using a single subscale for men (6 items) and the total FSFI score for women?

 This has been addressed in comment 2 above. 

4. Results: All tables must be presented in APA format

-If scales were used, one would expect Table 2 (a, b) to present frequencies/means of variables and not items. Why were the tables presented item by item? When do variables cease to be items and become variables for analysis? What was used in analysis - items in the questionnaire or variables?

 This has been addressed. All tables are in APA format Pages 11, 12,13,14,15,16,17,18

5. References do not follow any distinct style. Check PLoS ONE requirements for presenting references. The references have been revised 

Reviewer #3: 

Abstract:

- By taking into account the study period, sampling technique, data collection tools, and statistical methodologies (descriptive or inferential statistics), the authors needed to update the methods part of the abstract once more in light of their objectives.

 The methods section of the abstract has been revised. We added that ‘Statistical analyses were conducted using Stata release 15, separately for males and females. Frequencies to the FSFI and IIEF were estimated with their percentages. Categorical variables were described using frequencies and percentages while continuous variables were described using median (IQR). Two binary outcome variables “sexual dysfunction for men and sexual dysfunction for women” were generated. To investigate the association of sexual dysfunction with socio-demographic and psychosocial and psychiatric illness factors, a two-step procedure was adopted. During the first step, bivariate associations between each of the outcome variables and the independent variables were assessed using simple logistic regression models. In the second step, those socio-demographic, psychosocial and psychiatric illness factors that attained a level of significance of P ≤ 0.2 (liberal cut-off point) with bivariate analyses were included in the final multivariate logistic regression models that assessed for the factors that were independently significantly associated with sexual dysfunction. We were not able to conduct subgroup analyses by homo- and heterosexual respondents) 

- The authors must indicate the percentage with a 95% confidence interval for the prevalence of sexual disfunction for males and females, respectively.

 This has been incorporated Page 12

- In light of their findings, the authors ought to revise their recommendations.

 Although SD exists in both female and male patients with HIV, there need for specific interventions for the different genders. 

Introduction:

- The introduction section focuses primarily on the definition, nature, and determinants of sexual dysfunction, with little to no discussion of its prevalence. I should suggest that the authors review the global literature on the prevalence in order to briefly explain the epidemiology (the distribution) of sexual dysfunction.

 The section has been revised to include information on the global burden of SD 5

Methods:

-When was this study done? considering that the information came from a secondary source and that the patients with HIV/AIDS were chosen from medical records from what years?

- In addition to mentioning that this research was conducted as part of Prof Eugene Kinyanda’s Senior EDCTP Fellowship funded study entitled, ‘Clinical trials in HIV/AIDS in Africa: Should they routinely control for mental health factors?’, I believe the authors missed an important section of the materials and methods section, such as how the participants for the study were selected (sampling technique) and the determination of sample size, data collection tools and techniques, the operational definitions of some concepts, etc.

 Thank you for the observation. More details have been added on page 7 of the revised manuscript 7

- The authors should thoroughly discuss the statistical part of how categorical and continuous data are reported, respectively, and for variables in the final multivariate model, the authors should plan the adjusted odds ratio (AOR) along with the 95% confidence interval. When measuring the outcome variable, the authors needed to clearly categorize it into binary data by coding it.

 This has been done Page 12

- The authors had to identify at what level of significance the variables should be deemed significant in the final multivariate model, which also needed to be assessed and presented together with an AOR and a 95% confidence interval.

 This has been done on the multi variate model. Page 18

- Additionally, the authors had to disclose the precise categorization (their levels) of each independent variable measured in addition to stating the scales.

Results:

 This has been added Page 11

- I'm not sure why the authors chose to use the median rather than the mean when reporting the age.

 We chose the median over mean because the mean did not accurately reflect the central tendency 

- The authors had to specify what low and high socioeconomic status meant when classifying socioeconomic status based on the wealth index in the method section before reporting here.

 This has been added Page 7

- It is required to give the mean, plus or minus standard deviation, or the sub-total frequency of each domain of measurement scales when reporting sexual dysfunction in males and females. Additionally, for each outcome variable, the authors needed to indicate the 95% confidence interval.

 This was reported using prevalence and 95% confidence intervals. Page 12

- I'll advise you to place the subheading factors related to sexual dysfunction before presenting the tables labelled "bivariate analysis of related factors (not adjusted) and "multivariate model (tables 3 and 4).

 This has been done Pages 16 and 17

Discussion:

- The finding was well discussed by the authors. The authors were also required to expand on their discussion by including earlier prevalence research related to sexual dysfunction as well as further implications of the study for low-income nations like those in sub-Saharan Africa.

 This has been done Page 26 and 27

- Although it appears that the authors are talking about qualitative findings in their discussion, I believe that they should concentrate on quantitative results instead because they evaluated the study in a quantitative manner, unless they felt the need to include qualitative data.

Conclusion and recommendation:

 The study did not include any qualitative analyses – this was highlighted in the limitations 

- The authors should have provided a more thorough summary of the main findings and recommendations based on those findings, particularly for those stakeholders working on behavioural changes related to HIV/AIDS and for concerned policymakers.

At last, the authors produced some excellent work, but they should pay particular attention to the methods section because the majority of its components are missing. This has been included Page 26

---

## [Editor Report · Decision Letter 2]

6 Nov 2023

PONE-D-22-24725R2A ‘hidden problem’: Nature, prevalence and factors associated with sexual dysfunction in persons living with HIV/AIDS in UgandaPLOS ONE

Dear Dr. Rukundo,

Thank you for submitting your manuscript to PLOS ONE. After careful consideration, we feel that it has merit but does not fully meet PLOS ONE’s publication criteria as it currently stands. Therefore, we invite you to submit a revised version of the manuscript that addresses the points raised during the review process.

ACADEMIC EDITOR: Deal with citations (both in-text and reference page) in line with journal requirements.==============================

We look forward to receiving your revised manuscript.

Kind regards,

Habil Otanga, Ph.D

Academic Editor

PLOS ONE

Journal Requirements:

Additional Editor Comments:

Your work still suffers serious citation lapses. Go to the PLoS ONE webpage and find out how in-text and reference page citations are done.

---

## [Author Response · Author response to Decision Letter 2]

17 Nov 2023

The references have been revised and put in the Vancouver style. I hope they are now in the acceptable format. I have also corrected some punctuations and spelling mistakes. Thank you.

---

## [Editor Report · Decision Letter 3]

20 Nov 2023

A ‘hidden problem’: Nature, prevalence and factors associated with sexual dysfunction in persons living with HIV/AIDS in Uganda

PONE-D-22-24725R3

Dear Dr. Godfrey Zari Rukundo,

We’re pleased to inform you that your manuscript has been judged scientifically suitable for publication and will be formally accepted for publication once it meets all outstanding technical requirements.

Kind regards,

Habil Otanga, Ph.D

Academic Editor

PLOS ONE
---

## [Editor Report · Acceptance letter]

27 Nov 2023

PONE-D-22-24725R3 

A ‘hidden problem’: Nature, prevalence and factors associated with sexual dysfunction in persons living with HIV/AIDS in Uganda 

Dear Dr. Rukundo:

I'm pleased to inform you that your manuscript has been deemed suitable for publication in PLOS ONE. Congratulations! Your manuscript is now with our production department. 

Kind regards, 

on behalf of

Dr. Habil Otanga 

Academic Editor

PLOS ONE